# Nicotinamide Adenine Dinucleotide Phosphate Oxidases in Glucose Homeostasis and Diabetes-Related Endothelial Cell Dysfunction

**DOI:** 10.3390/cells10092315

**Published:** 2021-09-04

**Authors:** Oliver Ian Brown, Katherine Isabella Bridge, Mark Thomas Kearney

**Affiliations:** Leeds Institute of Cardiovascular and Metabolic Medicine, University of Leeds, Leeds LS2 9JT, UK; medoib@leeds.ac.uk (O.I.B.); k.bridge@leeds.ac.uk (K.I.B.)

**Keywords:** NADPH oxidases, NADH, NADPH, Nox, type 2 diabetes mellitus, endothelial function, oxidative stress, reactive oxygen species, atherosclerosis

## Abstract

Oxidative stress within the vascular endothelium, due to excess generation of reactive oxygen species (ROS), is thought to be fundamental to the initiation and progression of the cardiovascular complications of type 2 diabetes mellitus. The term ROS encompasses a variety of chemical species including superoxide anion (O_2_^•−^), hydroxyl radical (OH^−^) and hydrogen peroxide (H_2_O_2_). While constitutive generation of low concentrations of ROS are indispensable for normal cellular function, excess O_2_^•−^ can result in irreversible tissue damage. Excess ROS generation is catalysed by xanthine oxidase, uncoupled nitric oxide synthases, the mitochondrial electron transport chain and the nicotinamide adenine dinucleotide phosphate (NADPH) oxidases. Amongst enzymatic sources of O_2_^•−^ the Nox2 isoform of NADPH oxidase is thought to be critical to the oxidative stress found in type 2 diabetes mellitus. In contrast, the transcriptionally regulated Nox4 isoform, which generates H_2_O_2_, may fulfil a protective role and contribute to normal glucose homeostasis. This review describes the key roles of Nox2 and Nox4, as well as Nox1 and Nox5, in glucose homeostasis, endothelial function and oxidative stress, with a key focus on how they are regulated in health, and dysregulated in type 2 diabetes mellitus.

## 1. Introduction

Reactive oxygen species (ROS) are a by-product of oxygen metabolism [1]. The term ROS encompasses a variety of diverse chemical species including the superoxide anion (O_2_^•−^), hydroxyl radical (^•^OH) and hydrogen peroxide (H_2_O_2_). O_2_^•−^ and ^•^OH are extremely unstable, whereas H_2_O_2_ is uncharged, stable and can cross membranes freely, making it a more versatile signalling molecule [2,3]. Knowledge generated over the last three decades has revealed a diverse repertoire of actions for ROS including regulation of cell growth, migration, proliferation and metabolism [4]. ROS can be generated as a result of a number of different cellular processes, including: xanthine oxidase reactions, uncoupled nitric oxide synthases, the mitochondrial electron transport chain and by the nicotinamide adenine dinucleotide phosphate (NADPH) oxidases (Nox) [5].

The discovery of the Nox isoforms (Nox1–Nox5), whose only known function is to catalyse the synthesis of ROS [6,7], and the seminal finding that Nox4, which is transcriptionally regulated, generates H_2_O_2_, as opposed to superoxide, highlights the fact that ROS are not uniformly toxic, and may serve important physiological roles [8,9].

Type 2 diabetes mellitus is characterised by insulin resistance, inflammation, dyslipidaemia, dysglycaemia and intracellular nutrient overload. In endothelial cells (EC), this results in excessive generation of ROS, a situation often described as ‘oxidative stress’ or ‘endothelial dysfunction’ [1,10]. Large scale genetic studies support an emerging paradigm that puts oxidative stress at the centre of the pathophysiology of the insulin resistance associated with type 2 diabetes mellitus, and its associated cardiovascular complications. [11,12]. Consistent with this, insulin resistance, induced by a range of mechanisms in murine models of type 2 diabetes mellitus, leads to cytotoxic levels of O_2_^•−^, principally produced by the Nox2 isoform of NADPH oxidase (Nox2) [13,14,15,16]. Furthermore, it has been shown that expression of NADPH oxidase is increased within the endothelium of overweight and obese human subjects, [17] as well as patients with type 2 diabetes mellitus, resulting in an increase in EC generation of O_2_^•−^ [18]. The substantial dataset demonstrating the effect of oxidative stress in humans and preclinical models underpins its potential as a therapeutic target to mitigate the risks of accelerated cardiovascular disease in type 2 diabetes mellitus. However, a non-specific approach to reduce oxidative stress with various antioxidants has failed to improve clinical outcomes, with meta-analyses of clinical trials showing that antioxidants may even increase mortality [19].

The aim of this review is to describe the role of Nox in glucose homeostasis, endothelial function and oxidative stress, with a key focus on how they are regulated in health and dysregulated in type 2 diabetes mellitus.

## 2. Formation of ROS

Synthesis of ROS is catalysed by xanthine oxidase, uncoupled nitric oxide synthase (NOS), the mitochondrial electron transport chain and the NADPH oxidases [7,20,21,22].

Xanthine oxidase, a form of xanthine oxidoreductase, catalyses the oxidation of hypoxanthine to xanthine, and of xanthine to uric acid—the last two steps of purine catabolism [23]. Transfer of electrons during these two steps, using molybdopterin, two iron-sulphur centres and flavin adenine dinucleotide (FAD) as co-factors, leads to the formation of H_2_O_2_ and O_2_^•−^ from O_2_ [23].

Nitric oxide synthases (NOS) catalyse the transformation of l-arginine, O_2_ and NADPH derived electrons to nitric oxide (NO) [24]. Tetrahydrobiopterin (BH_4_) is a crucial co-factor in this reaction, acting as a rapid auxiliary electron donor, stabilising the process of electron transfer. When BH_4_ availability is reduced, the electron transfer process is too slow and NOS catalyses the reduction of O_2_ to O_2_^•−^, rather than the formation of NO—a phenomenon known as NOS uncoupling [24]. Furthermore, BH_4_ is rapidly degraded by O_2_^•−^, leading to a vicious cycle, where oxidative stress worsens BH_4_ availability, thus further potentiating the problem [25].

NO is a critical anti-oxidant and anti-inflammatory signalling radical that has favourable effects on inflammation, thrombosis, EC death, vascular tone, and angiogenesis. NO is synthesized in EC principally by endothelial NOS (eNOS) [26]. In the short term, the bioavailability and actions of NO are modulated by ROS, specifically O_2_^•−^ [27]. In the longer term, eNOS activity can be regulated at the transcriptional, post-transcriptional and post-translational levels [26,28].

Aside from NO, EC function is dependent on the action of prostanoids—products of arachidonate metabolism via the action of cyclooxygenases (COX) [29]. Several types of prostanoids exist including: prostaglandin D2 (PGD_2_), prostaglandin I2 (PG_I2_), prostaglandins F2α (PGF_2α_), prostaglandin E2 (PGE_2_), and thromboxane A2 (TXA_2_). Prostaglandin H2 (PGH_2_) is an important intermediary step in their synthesis and is converted into specific prostanoids by various prostanoid synthases [29]. PGE_2_ and PG_I2_ act as vasodilators whilst TXA_2_ and PGF_2α_ induce vasoconstriction [29]. PGD_2_ has variable effects on vascular tone depending on site and concentration [30]. Importantly, the activity of COX, prostanoid synthases and therefore prostanoid production is regulated by ROS. COX stimulates production of the vasoconstricting PGE_2_ and TXA_2_, which in turn stimulates production of ROS by Nox, O_2_^•−^ produced by Nox then activates COX leading to a vicious cycle of self-perpetuating endothelial dysfunction [31].

ROS are generated in the mitochondria due to mitochondrial inefficiency during the electron transport chain, with mitochondrial ROS being the major intracellular source of ROS under physiological conditions. As electrons pass sequentially along mitochondrial transmembrane protein complexes (I, II and III), approximately 0.2–2% of electrons leak out of the transport chain. These electrons react with O_2_ within the mitochondria to form O_2_^•−^ and H_2_O_2_ [22].

Nox are a family of membrane-spanning protein complexes which generate extracellular ROS by the transfer of electrons from NADPH to oxygen, resulting in the formation of O_2_^•−^ and H_2_O_2_ [32]. At least seven isoforms of the catalytic component of Nox exist (Nox1, Nox2, Nox3, Nox4, Nox5, DUOX1 and DUOX2) and their unifying function is that they only serve to produce ROS [5]. Distribution of specific Nox isoforms within the body is highly variable in terms of tissue and cell type. However, only Nox1, Nox2, Nox4 and Nox5 are expressed within EC and will therefore be the focus of this review.

## 3. ROS and Endothelial Cell Function

Intracellular production of ROS in the endothelium occurs under physiological as well as pathophysiological conditions. H_2_O_2_ is small, stable, and non-polar and is thus able to freely diffuse across membranes, playing a vital role in the activation of a number of cell signalling pathways. H_2_O_2_ stimulates transcription of a number of different signalling proteins including vascular endothelial growth factor (VEGF) and protein C-ets-1 which promote EC proliferation, tube formation and angiogenesis [30]. The relationship of H_2_O_2_ with oxidative stress is thought to be bell-shaped [4]. Low concentrations of H_2_O_2_ increase production of NO by upregulating expression of eNOS, and by direct activation of cyclic guanosine monophosphate (cGMP) through activation of protein kinase-G Iα, inducing thiol oxidation and subsequent eNOS dimerisation [33,34]. In addition, H_2_O_2_ increases phosphorylation and activation of the insulin receptor (IR) by oxidising cystine residues, and can activate Akt and eNOS, thus amplifying the insulin signalling cascade [35,36]. At high concentrations, H_2_O_2_ increases cell surface expression of intercellular adhesion molecule-1 (ICAM-1), platelet activating factor (PAF), P-selectin and monocyte chemoattractant protein-1 (MCP-1). These proinflammatory proteins mediate neutrophil adhesion to the endothelium and lead to accelerated atherosclerosis [37].

## 4. The Role of Nox in ROS Production

Human ECs express four Nox isoforms; Nox1, Nox2, Nox4 and Nox5. Nox1, Nox2 and Nox5 promote endothelial dysfunction, inflammation and apoptosis in the vessel wall through the generation of superoxide. All Nox isoforms are unified in their structure by the presence of a catalytic six transmembrane domain protein complex (see Figure 1A) [6]. The N-terminal transmembrane domain of this protein complex binds two heme groups, and the C-terminal domain binds to flavin adenine dinucleotide (FAD) which in turn binds with NADPH to form the Nox complex [6]. Furthermore, all Nox isoforms (with the exception of Nox5) are reliant on p22phox, a membrane anchored subunit for their activity (see Figure 1A) [7]. However, there are significant differences in the regulatory cytosolic subunits required for each Nox isoform to be active.

Nox1 and Nox2 are close structural homologs. Nox1 is localised in the peri-nuclear cytoplasmic skeleton, endoplasmic reticulum, caveolae and nuclear membrane whilst Nox2 is located in the plasma membrane, endoplasmic reticulum and the perinuclear cytoplasmic skeleton [6,7]. Nox1 and Nox2 require similarly acting regulatory cytosolic subunits to be active: Nox1 requires the activation of NOXA1, NOXO1, p40phox and Rac whilst Nox2 requires the activation ofp47phox (a functional homolog of NOXO1), p40phox, p67phox (a functional homolog of NOXA1), and Rac (see Figure 1A).

Nox4 has a similar membrane bound structure to Nox1/Nox2, and is located within the endoplasmic reticulum, plasma and nuclear membranes. However, unlike Nox1 and Nox2, Nox4 does not require cytosolic subunits for its activity and appears to be constitutively active [7,32]. Nox5 does not have any regulatory cytosolic subunits and is unique in that it contains a calmodulin-like domain and is activated by changes in intracellular Ca^2+^ [6,38].

Nox1, Nox2 and Nox5 generate superoxide anions by reducing O_2_ to O_2_^•−^, by transferring electrons from NADPH in the cytosol to oxygen on the extracellular surface (see Figure 1B).

The mechanism of O_2_^•−^ related tissue damage is largely twofold. Firstly, O_2_^•−^ is highly reactive with transition metal complexes, disrupting mitochondrial function [39]. Secondly, O_2_^•−^ rapidly inactivates NO. Not only does this reduce the bioavailability of NO, but the reaction between O_2_^•−^ and NO produces peroxynitrite (ONOO⁻), a potent oxidant [27]. ONOO⁻ can cause oxidative damage directly through reactions with key protein moieties such as thiols, iron/sulphur centres and through tyrosine nitration. Furthermore, ONOO⁻ oxidises BH_4_, an essential cofactor of eNOS, which leads to oxygen reduction uncoupling from NO synthesis, and increased formation of O_2_^•−^ [40]. ONOO⁻ can also cause oxidative damage indirectly, by decomposition into other reactive radicals such as ^•^OH [27,41]. ^•^OH can react with DNA, proteins and lipids within the cell leading to significant oxidative damage [37]. O_2_^•−^ is rapidly scavenged by superoxide dismutase (SOD) into H_2_O_2,_ negating its oxidising effects. We have previously demonstrated, through EC specific inactivation of Shc homology 2-containing inositol 5′ phosphatase-2 (SHIP2), a lipid phosphatase that inhibits insulin signalling downstream of PI3K, that Nox2-derived endothelial ROS production is dependent on the P13K/Akt pathway. [42]

Unlike other Nox isoforms, the O_2_^•−^ produced by Nox4 is rapidly converted to H_2_O_2_, rendering the production of O_2_^•−^ by Nox4 essentially undetectable. [32]

## 5. Regulation of Nox

The regulation of Nox is summarised in Figure 2.

### 5.1. Transcriptional Regulation of Nox

In humans, the genes encoding Nox1 and Nox2 can be found on the X chromosome, whilst Nox4 and Nox5 can be found on chromosome 11 and 15 respectively [7,43]. A wide variety of protein signalling cascades, hormones and cytokines including mitogen activated protein kinase (MAPK), interferon gamma (IFNγ), tumour necrosis factor alpha (TNF-α), transforming growth factor beta (TGF-β), and angiotensin II act to increase Nox through transcription factors [44,45,46,47]. Nox expression is negatively regulated by JunD [48]. Initial work identified peroxisome proliferator-activated receptor gamma (PPAR-γ) as a negative regulator of Nox, [49] however recent work has suggested Nox-dependent generation of O_2_^•−^ is dependent on other types of PPAR (PPARα and PPARβ/δ) [50]. Nox1 expression is upregulated by a number of transcription/growth factors including: platelet derived growth factor (PDGF), activating transcription factor-1 (ATF-1), signal transducer and activator of transcription protein (STAT), AP-1 and NF-κB [51,52]. In addition, proinflammatory cytokines (interleukin-1 and IFNγ); angiotensin II and PGF2α increased Nox1 activity [53,54]. Nox2 promoter activity is increased by, but not limited to: transcription factor PU.1, E74-like factor 1 (Elf-1), interferon regulatory factors 1 (IRF-1), interferon consensus sequence binding protein (ICSB), activator protein 1 (AP-1), STAT, CCAAT/enhancer-binding proteins (C/EBP) and nuclear factor kappa-light-chain-enhancer of activated B cells (NF-κB) [45,50,55,56].

Nox4 is constitutively active and is regulated by large number of transcription factors that promote its expression including: nuclear factor erythroid 2-related factor 2(Nrf2), Smad2, Smad3, E2F, hypoxia-inducible factor 1-alpha (HIF1α), C/EBP, AP-1 and NF-κB [43,45]. Angiotensin II has been shown to more potently activate Nox2 than Nox4 [57,58]. In specific relation to Nox4, PDGF also reduces Nox4 messenger RNA (mRNA) [43]. Protein phosphatase 2 (PP2A) downregulates Nox2 expression, with inhibition of PP2A in rats being shown to exacerbate non-alcoholic steatohepatitis (a disease associated with metabolic syndrome and type 2 diabetes mellitus) [59]. In addition, endoplasmic reticulum stress upregulates Nox1–5 expression [60,61].

The promotor region of the Nox5 gene has binding sites for multiple transcription factors including: AP-1, NF-κB, STAT and C/EBP [56,62]. In addition, expression of Nox5 mRNA is significantly increased by angiotensin II, endothelin-1 and TNF-α [62].

### 5.2. Epigenetic Regulation of Nox

Epigenetic mechanisms, including the activity of histone methylators/deacetylases/acetylators, DNA methylation and microRNAs (miRs), have been postulated as playing a significant role in the regulation of Nox expression [63].

Histone deacetylases (HDACs) are a class of enzymes which act directly on Nox transcription by removing acetyl groups from histone proteins, reducing chromatin relaxation and suppressing gene transcription [64]. HDACs also have some activity on non-histone proteins (including transcription factors like AP-1 and NF-κB), indirectly suppressing gene transcription [63]. Although, and what may appear somewhat paradoxical, inhibition of HDACs reduces expression of Nox [65,66]. This may be in part due to hyperacetylation at Nox promotor regions, blocking the action of RNA polymerase II and c-jun (a key constituent protein unit found in AP-1) when HDAC is inhibited [64,66,67]. DNA methylation and histone modification may play a key role in the regulation of Nox [48]. DNA at the Rac1 promoter region (a key subunit of Nox1/2) undergoes methylation in response to hyperglycaemia within retinal ECs and is associated with increased Nox expression and activity [68]. Furthermore, Hussain et al. elegantly showed within a diabetic murine model that the downregulation of JunD (a negative regulator of Nox) was mediated by hypermethylation and post-translational histone methylation within the promoter region of JunD [48]. Similarly, increased histone acetylation (HAT) within the promoter region of Nox5 (with an associated increased Nox5 activity) has been found in human atherosclerotic tissue samples [69]. Acetylation of non-histone proteins like transcription factors, may also have a key role to play in the regulation of Nox [70].

miRs are a large class of non-coding RNA that play a critical role at the post transcriptional level, by binding to complementary mRNAs at specific sequences located within the 3′UTR of target mRNA, leading to transcriptional inhibition or degradation [71] miRs are found to be dysregulated in a range of disorders including cardiovascular disease, diabetes and cancer [72,73,74,75,76,77]. There is a growing body of evidence that an abundance of miRs play an important role in regulation of Nox (see recent review by Wlodarski et al. [78]). Of particular interest is miR-25.

A number of studies have investigated the relationship between miR-25 and Nox4, although with conflicting results. Early studies of miR-25 in a rat diabetes model demonstrated miR-25 downregulated Nox4 expression [79,80]. More recently, Liu et al. demonstrated that treating diabetic mice with an miR-25 antagomir reduced Nox4 expression; whilst inhibition of miR-25 in healthy mice led to increased blood pressure and renal dysfunction [81]. Liu et al. also observed that miR-25 was decreased in plasma from diabetic patients, and that upregulation of miR-25 in obese mice reversed diabetes induced nephropathy and reduced blood pressure by inhibiting the renin-angiotensin aldosterone system [81]. Despite clear links between miR-25 and Nox4 expression in both models of diabetes and human studies, inhibition of miR-25 in patients with heart failure has no effect on myocardial Nox4 expression—suggesting the role of miR-25 in the regulation of Nox4 may be tissue and context specific [82].

In EC specifically, miR-146a and miR-92b have both been shown to reduce Nox4 expression, [83,84] whilst other miR that have been shown to downregulate Nox4 expression include miR-363-3p, miR-182, miR-99, miR-590, miR-423-5p, miR-125b, miR-29a-5p, miR-29c-3p, and miR-132-3p and miR-17 [85,86,87,88,89,90,91,92].

High-throughput screening of miRs targeting Nox2 has identified miR-106b, miR-148b, miR-204 and miR-448-3p as important negative regulators of Nox2 at a gene and protein level [93,94]. Other negative regulators of Nox2 expression reported in the literature include: miR-320, miR-27b, miR-99, miR-125b, miR-18a, miR-210, miR-126a-5p, miR-29c, miR-17 and miR-34a [87,90,91,92,95,96,97,98,99,100].

Compared to Nox2 and Nox4 there is a smaller body of published research investigating the regulation of Nox1 and Nox5 by miR within the endothelium. However, miR-1264, miR-298-5p and miR-217 have been shown to reduce expression of Nox1 in vascular smooth muscle cells and macrophages [101,102,103]. miR-15a-3p has been shown to reduce Nox5 expression within human ECs [104]; whilst miR-465, miR-4321 and miR-4270 have been reported as being negative regulators of Nox5 in the literature [105,106].

## 6. Oxidative Stress in Type 2 Diabetes Mellitus

Type 2 diabetes mellitus is a disorder of metabolism and fuel homeostasis characterised by insulin resistance, with resultant hyperglycaemia [107].

Insulin resistance is associated with endothelial dysfunction [108]. Insulin stimulates the production of NO by acting on the insulin receptor and insulin-like growth factor 1 receptor, leading to a rapid, dose-dependent production of NO via the PI3K/Akt signalling cascade [109,110,111]. Akt induces phosphorylation and activation of eNOS at serine 1177, which in turn stimulates the transfer of electrons from NADPH resulting in the conversion of l-arginine to l-citrulline and the formation of NO (Figure 3) [111].

Hyperglycaemia can directly increase ROS formation by enhancing the mitochondrial electron transport chain and eNOS uncoupling [112,113]. Furthermore, hyperglycaemia also stimulates Nox to produce ROS, by increasing synthesis of di-acylglycerol (DAG) and activation of PKC [112]. In addition, nutrient excess in type 2 diabetes mellitus can overload the protein folding capacity of the endoplasmic reticulum (ER) resulting in the formation of H_2_O_2_ and O_2_^•−^ [114,115]. Hyperglycaemia also leads to the formation of advanced glycation end products (AGEs); proteins which have undergone glycosylation and which instigate EC dysfunction [116]. In murine models of diabetes, AGE also increase production of ROS by mitochondria and Nox [117,118]. Methylglyoxal (MGO), a highly reactive dicarbonyl compound, is a major precursor of AGE formed during glucose and fatty acid metabolism [119]. Moreover, production of MGO is regulated by Nox [120].

In addition to activation of the PI3K-Akt pathway via tyrosine kinase, insulin and IGF-1 can activate several MAPKs, including c-Jun N-terminal (JNK), extracellular signal regulated kinase 1/2 (ERK1/2) and IκB kinase (IKK). MAPKs are activated through a series of phosphorylative protein reactions dependent on the interaction of Ras with Shc adaptor protein [35]. ERK1/2, JNK and IKK can phosphorylate IRS-1 and interrupt the PI3K-Akt pathway, reducing eNOS synthesis and production of NO [121,122]. MAPK are inactivated by MAPK phosphatases [123], the action of which is inhibited by ROS [124].

Insulin resistance is associated with chronic activation of pro-inflammatory signalling pathways [125]. The mechanism underpinning this association is thought to be mediated by variety of cytokines including TNFα, interleukin (IL)-1β, IL-10 and IFNγ; all of which are found at high levels within macrophages in adipose tissue [126]. TNFα and IL-1β activate both the IKK/NF-κB and JNK signalling cascade leading to inhibition of downstream insulin signalling by serine phosphorylation of IRS-1 [126].

The MAPK pathway is thought to be pivotal in the pathophysiology of endothelial dysfunction in type 2 diabetes mellitus. Serum insulin levels increase in response to peripheral insulin resistance; however, as insulin is unable to activate the PI3K-Akt pathway it increases activation of the MAPK pathway leading to reduced production of eNOS. Moreover, increasing quantities of ROS from hyperglycaemia and hyperinsulinaemia enhance the activity of the MAPK pathway. Finally, MAPK, IFNγ, TNF-α and transforming growth factor (TGF)-β all upregulate expression of Nox through the action of numerous transcription factors, increasing the production of O_2_^•−^ and leading to a vicious, perpetuating cycle culminating in reduced production of eNOS and endothelial dysfunction [44,45,47,54].

## 7. Nox1 in Type 2 Diabetes Mellitus Related Endothelial Dysfunction

Human aortic endothelial cells exposed to hyperglycaemia showed increased expression of Nox1, oxidative stress and proinflammatory markers in a Nox1-siRNA reversible manner [127]. In addition, in a murine model of atherosclerosis, deletion of *NOX1* had a profound anti-atherosclerotic effect. Thus, the study suggests, Nox1-dependent oxidative stress is a promising target for diabetic vasculopathy, including atherosclerosis [127].

Excess ROS derived from Nox1 and Nox4 have been shown to contribute to diabetic retinopathy, an end-stage disease of diabetes mellitus, which threatens vision as a result of increased vascular permeability and neovascularisation [128]. Inhibition of Nox1/4 in bovine endothelial cells reduces the high glucose induced oxidative stress, angiogenic and inflammatory markers, indicated the potential of Nox1 and Nox4 inhibition in reducing the vision damage associated with diabetic retinopathy [129].

The importance of Nox1 in diabetic nephropathy, another end-stage disease of diabetes mellitus, has been demonstrated using a rodent model of genetic obesity. In obese zucker rats, Nox1-derived O_2_^•−^ was the main cause of renal endothelial dysfunction, highlighting the need to identify and target the specific Nox isoforms involved in tissue-specific metabolic disease [130].

## 8. Nox2 in Type 2 Diabetes Mellitus-Related Endothelial Dysfunction

In vivo studies of whole body and EC specific insulin resistance using gene-modified mice have offered fascinating insights into the role of Nox2 in the pathophysiology of type 2 diabetes mellitus and its cardiovascular complications [16,131,132,133,134,135].

High fat feeding, with resultant whole body insulin resistance and glucose intolerance, is an established murine model for type 2 diabetes mellitus, and has been shown to increase Nox2 protein expression and superoxide production [16]. Further to this, although *NOX2*-KO mice fed a high fat diet still demonstrated insulin resistance compared to *NOX2*-KO mice on a standard diet, the magnitude of insulin resistance was significantly smaller, and the production of O_2_^•−^ was no longer increased, implying that Nox2 has a pivotal role in the insulin resistance and ROS production associated with type 2 diabetes [136]. Akt phosphorylation and glucose transporter 4 (GLUT4) translocation were decreased in skeletal muscle of wild type mice fed a high fat diet but this effect was not seen in *NOX2*-KO mice [136]. This suggests that deletion of *NOX2* leads to increased sensitivity to insulin, whilst protecting against the detrimental effects of high fat diet through Nox2-dependent regulation of Akt phosphorylation and GLUT4 expression [136].

The effects of EC specific insulin resistance have been assessed using a transgenic mouse with endothelial overexpression of an insulin resistant mutant IR (ESMIRO). Alongside their disrupted EC specific insulin signalling, ESMIRO mice demonstrated increased expression of EC Nox2 and Nox4 [13]. Subsequent work has gone on to assess the effects of inhibiting Nox2 in this model and in mice with insulin resistance due to haploinsufficiency of the IR at the whole-body level (IR^+/−^). Nox2 dependent O_2_^•−^ production was significantly increased in pulmonary EC from ESMIRO and IR^+/−^ mice [16]. Furthermore, knockdown of *NOX2* in pulmonary EC of ESMIRO and IR^+/−^ mice led to significant reduction of O_2_^•−^, confirming Nox2 as an important intermediary in the ROS-induced endothelial dysfunction associated with insulin resistance [16]. By crossing the ESMIRO mice with *NOX2* holoinsufficient mice (Nox2^y/−^), we were able to study the effect of chronic Nox2 deficiency in the setting of insulin resistance. Basal production of O_2_^•−^ within pulmonary EC of ESMIRO/Nox2^y/−^ mice was reduced compared to that of ESMIRO mice alone, highlighting that Nox2 is a vital intermediary in the production of ROS in insulin resistance, and confirming that there is no adequate compensatory mechanism for increasing ROS production in the setting of insulin resistance if Nox2 is deficient [16].

Interestingly, in the setting of enhanced EC insulin sensitivity, we have been able to demonstrate increased Nox2 dependent O_2_^•−^ release and reduced NO bioavailability. In fact, genetically enhancing endothelial insulin sensitivity results in a contradictory decrease in endothelial function, a change that is driven by an increase in Nox2-dependent oxidative stress and a PYK2-dependent reduction in eNOS activity [15]. This adds further weight to the idea that relationship between ROS production and EC function is likely bell-shaped.

Given that endothelial dysfunction increases the likelihood of atherosclerosis development, we have investigated the relationship between Nox2 and atherosclerosis development using apolipoprotein E-deficient ESMIRO mice (ESMIRO/ApoE^−/−^) with genetic and pharmacological inhibition of Nox2 [18]. ESMIRO/ApoE^−/−^ mice with germline knockdown of *NOX2* (ESMIRO/ApoE^−/−^/Nox2^−/y^) had reduced O_2_^•−^ generation within pulmonary EC and improved aortic relaxation to acetylcholine compared to their ESMIRO/ApoE^−/−^/Nox2^+/y^ littermates [18]. In addition, ESMIRO/ApoE^−/−^/Nox2^−/y^ mice had increased thoraco-abdominal aortic lipid deposition, and multiple foci of atherosclerosis associated elastin fragmentation in the aortic sinus. These changes were accompanied by increased aortic expression of ICAM-1 and VCAM-1 [18]. However, inhibition of Nox2 with gp91dstat, whilst also reducing pulmonary EC O_2_^•−^ reduced aortic atherosclerosis and elastin fragmentation in ESMIRO/ApoE^−/−^ mice [18] Once again, it is clear that the relationship between insulin resistance and Nox2 dependent O_2_^•−^ release is far from linear. The divergent effects of genetic versus pharmacological inhibition of Nox2 are likely to be complex and may be related to the duration of Nox2 inhibition.

Within the microvasculature, Nox2-derived ROS is fundamental in the development of diabetes induced retinal inflammation and diabetic retinopathy [137,138]. Deletion of the *NOX2* gene reduces ROS production, VEGF and ICAM-1 expression, and the breakdown of the blood-retinal barrier [137,138]. Furthermore, diabetes increases Nox2 generated ROS within bovine retinal premature EC, inducing EC senescence by increases in expression and activity of arginase-1 (which decreases NO bioavailability). Inhibition of Nox2 in bovine retinal EC and deletion of *NOX2* in mice ameliorates premature EC senescence by limiting the activity of arginase-1 [139].

## 9. Nox4 and Endothelial Function

Initially, Nox4 was thought to also play a pathological role within the endothelium [140]. Studies using siRNA within cultured EC implicated Nox4 as being responsible for causing oxidative damage within the endothelium, reducing EC replicative potential and causing endothelial dysfunction [140]. Wang et al. demonstrated that hyperglycaemia increased protein expression of Nox4 within human aortic EC; with associated increased production of ROS, IL-6 and IL-8 [83]. Nox4 activity has also been shown to be increased in the serum of patients with type 2 diabetes mellitus [141]. More recently, however, in a unique murine model of EC insulin resistance with whole body insulin sensitivity, we demonstrated reduced EC expression of Nox2 and miR-25, but increased expression of Nox4 and an associated increased production of H_2_O_2_ [142]. Furthermore, transgenic mice with endothelial targeted overexpression of *NOX4* had increased H_2_O_2_ production and H_2_O_2_-induced hyperpolarization (without affecting endothelial NO bioactivity) within their coronary microvascular ECs, resulting in significantly greater acetylcholine- or histamine-induced vasodilatation compared to their wild type littermates [143].

Moreover, data from KO mice suggest that Nox4 might offer further protective benefits within the endothelium by promoting angiogenesis [144,145]. EC migration and proliferation is inhibited when the *NOX4* gene is silenced; whilst EC with overexpression of *NOX4* have increased angiogenesis in vitro [146,147]. Deletion of *NOX4* in mice reduces H_2_O_2_ production and inhibits EC tube formation. Moreover, the addition of low concentrations of H_2_O_2_ restores EC tube formation—highlighting the important role of Nox4 derived H_2_O_2_ in endothelial function [145]. Addition of cord blood-derived endothelial colony-forming cells, which have overexpression of *NOX4*, to an ischaemic murine hindlimb model upregulates several angiogenic factors and promotes post-ischaemic revascularisation [148]. Nox4 also appears to have anti-atherosclerotic properties [149]. Deletion of *NOX4* in ApoE^−/−^ mice fed a high fat diet accelerated atherosclerosis development and resulted in greater disease development after partial carotid artery ligation [149]. Furthermore, genetic deletion of *NOX4* in a low-density lipoprotein receptor (Ldlr) knockout mouse model reduced the vasodilatory capacity of the endothelium and increased atherosclerotic plaque burden, as a result of reduced H_2_O_2_ production [150].

Physiologically, Nox4 appears to play a significant role in regulating the metabolic and vascular adaptions needed during exercise [151,152]. Nox4-deficient mice had impaired glucose and fatty acid oxidation in response to acute exercise. Furthermore, following a chronic exercise regime, *NOX4*-deficient mice had worse exercise capacity (measured by a reduced time to exhaustion) and reduced activity of skeletal muscle citrate synthase and beta-hydroxyacyl-coA dehydrogrenase (important enzymes involved in mitochondrial metabolism) compared to their wild-type litter mates [151]. Access to an exercise wheel prevented endothelial dysfunction in wild-type, but not Nox4^−/−^ deficient mice fed a high fat diet [152]. Exercise of wild type mice resulted in increased H_2_O_2_ release in the aorta with increased phosphorylation of Akt and activation of eNOS, whereas both H_2_O_2_ release and phosphorylation of Akt were reduced in aortas of Nox4^−/−^ mice. In addition, knockout of *NOX4* prevented the normal exercise-induced increase in mitochondrial capacity as measured by exercise-induced citrate synthase activity and mitochondria mass [152]. This all suggests that Nox4 has a key mechanistic role in modulating the protective effects of physical activity on endothelial function.

## 10. Nox5 in Type 2 Diabetes Mellitus Related Endothelial Dysfunction

Nox5 is expressed in human ECs and is of growing interest in endothelial disease/dysfunction, due to its promotion of EC proliferation [153]. It is absent from rodents, and thus is not easily studied in vivo, despite being expressed by humans and other higher mammals [154]. There does, however, seem to be emerging evidence that Nox5 plays a critical role in a range of vascular diseases, including those associated with diabetes (review in [155]).

A recent sophisticated study, in which transgenic mice with inducible human Nox5 expressed in ECs were generated, demonstrated increased retinal vascular permeability and neovascularisation compared with their wild-type litter mates. In bovine retinal ECs, which express Nox5 (alongside Nox1 and 4), Nox5-siRNA also reduced high glucose induced oxidative stress, angiogenic and inflammatory markers, adding Nox5 inhibition to the potential list of therapeutic targets for reducing the visual damage associated with diabetic retinopathy [129].

A similar murine model, with increased expression of human Nox5 in the endothelium, has shown age-related severe systolic hypertension, and impaired endothelium-dependent vasodilation, as a result of Nox5-induced uncoupling of eNOS [156]. Further work within a conditional endothelial *NOX5* knock-in mouse, has identified a role for Nox5 in the response to ischaemia as a result of myocardial infarction, with evidence that Nox5-derived ROS may modulate the COX-2 and PGE_2_ axis in EC. [157].

## 11. Therapeutic Targeting of Nox

Initially it was thought that natural antioxidants (such as vitamin C, vitamin E, coenzyme Q amongst many others) may have a therapeutic role in reducing oxidative stress by targeting Nox [158]. These antioxidants have a range of actions and target a variety of subunits within the Nox complex [158]. However, the results of large-scale clinical trials using these antioxidants have been disappointing, showing no significant mortality benefit and for some agents (like vitamin A, vitamin E and beta-carotene) a tendency to cause harm [159].

More recently, specific Nox inhibitors have been developed, the full details of which are outside the scope of this review (see review in [160]). Problems of non-selectivity remain for many of these agents, due to their mechanism of action targeting protein subunits shared by several Nox isoforms [160]. NOX2ds-tat peptide is an exception to this rule, which binds to the p47phox subunit and is selective for Nox2. However, the low bioavailability of peptides limits its use within in vivo studies and as a therapeutic agent in humans [161].

GKT137831 (a dual Nox1/Nox4 inhibitor) was another Nox inhibitor of interest. [162] However, a clinical trial of GKT137831 vs. placebo in patients with type 2 diabetes mellitus related kidney disease and albuminuria failed to meet its primary endpoint of reduction in albuminuria, (https://www.businesswire.com/news/home/20150909005080/en/Genkyotex-Announces-Top-Line-Results-of-Phase-2-Clinical-Program (accessed on 31 August 2021)) the results of which have not been fully published. A phase II trial of GKT137831 in T1DM is ongoing [163].

Targeting the epigenetic machinery which regulate Nox transcription may provide a future avenue for therapeutic research [67,164]. Drugs inhibiting specific miRs involved in diabetes and endothelial dysfunction are being developed. These drugs are delivered using a variety of methods including: nanoparticles, dendrimers, aptamers or within liposomes [164]. Aptamers (small oligonucleotides which bind to a specific target molecule) are of specific interest as they have the essential ability to target a particular miRNA within a specific cell type [164]. Phase 2 trials of miR inhibitors are ongoing, albeit outside the field of diabetes. However, research in this area is still in its early stages and no therapeutic agent targeting miR within this field is currently close to coming to market [165].

## 12. Conclusions

Oxidative stress driven by ROS is a fundamental process in the pathogenesis of type 2 diabetes mellitus. Whilst generation of low concentrations of ROS are indispensable for normal cellular function, excess O_2_^•−^ potentiates endothelial dysfunction and progression of the cardiovascular complications of type 2 diabetes mellitus. Nox1, Nox2 and Nox5 generate O_2_^•−^ and are ubiquitously harmful; whilst Nox4 generates H_2_O_2_—which plays a more physiological role as a key signalling molecule. Whilst all Nox isoforms may be attractive targets for the development of novel therapeutic agents, it is clear that Nox expression is complex and likely tissue specific. Careful thought is therefore needed when devising new treatments targeting Nox, and its downstream effects. Manipulation of miR may overcome some of these issues in drug development. However, research in this area is still in its relative infancy.

## Figures and Tables

**Figure 1 cells-10-02315-f001:**
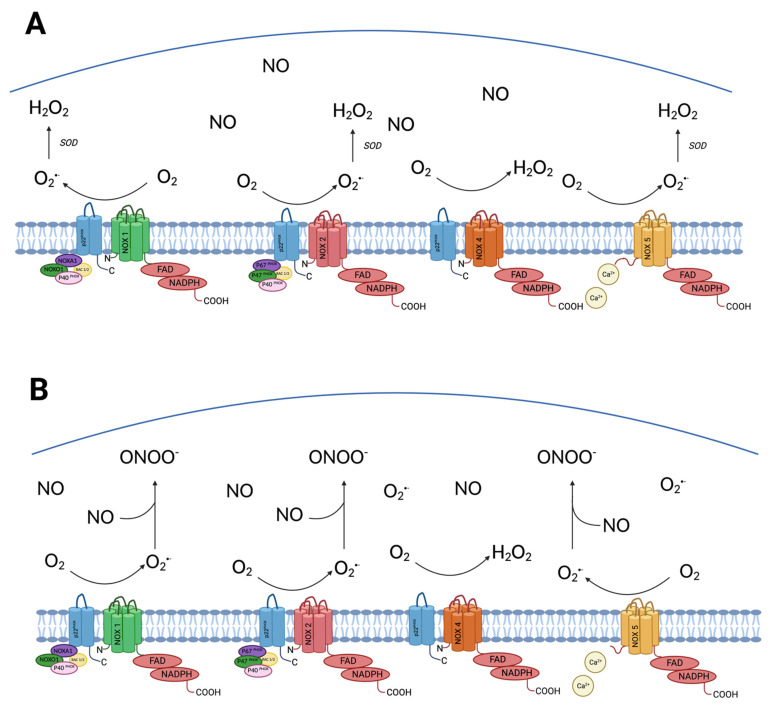
Panel (**A**) Structure and function of Nox1–5 isoforms of NADPH oxidase and their function in health: Nox1–5 all contain a catalytic six transmembrane domain protein complex, the N-terminal transmembrane domain of this protein complex binds two heme groups, and the C-terminal domain binds to FAD which in turn binds with NADPH to form the NADPH oxidase complex. All Nox isoforms (except Nox5) are reliant on p22phox, a membrane anchored subunit for their activity. Differences exist amongst the regulatory cytosolic subunits required for each Nox isoform. Nox1 requires activation of regulatory subunits NOXA1, NOXO1, p40phox and Rac whilst Nox2 (also called gp91phox) requires the activation of regulatory subunits p47phox, p40phox, and Rac to be active. Nox5 does not require any regulatory cytosolic subunits and is unique in that it contains a calmodulin-like domain and is activated by changes in intracellular Ca^2+^. Nox1, 2 and 5 isoforms of NADPH oxidase complex reduces oxygen to produce the O_2_^•−^ which is dismutated to H_2_O_2_ by SOD. Dismutation of O_2_^•−^ increases the bioavailability of NO. Nox4 does not require cytosolic subunits for its activity and appears to be constitutively active. Unlike all other Nox isoforms, the O_2_^•−^ produced by Nox4 is rapidly converted to H_2_O_2_ so production of O_2_^•−^ by Nox4 is essentially undetectable. Panel (**B**) The role of Nox1–5 in diabetes related endothelial dysfunction. The expression and activity of Nox1, 2 and 5 is increased in disease states like diabetes. This leads to increased production of O_2_^•−^ which combines with NO to form ONOO⁻, a very potent oxidiser and reduces the bioavailability of NO resulting in endothelial dysfunction. Abbreviations: flavin adenine dinucleotide (FAD), hydrogen peroxide (H_2_O_2_) nicotinamide adenine dinucleotide phosphate (NADPH), nitric oxide (NO) peroxynitrite (ONOO⁻), superoxide dismutase (SOD). Created with BioRender.com (accessed on 31 August 2021).

**Figure 2 cells-10-02315-f002:**
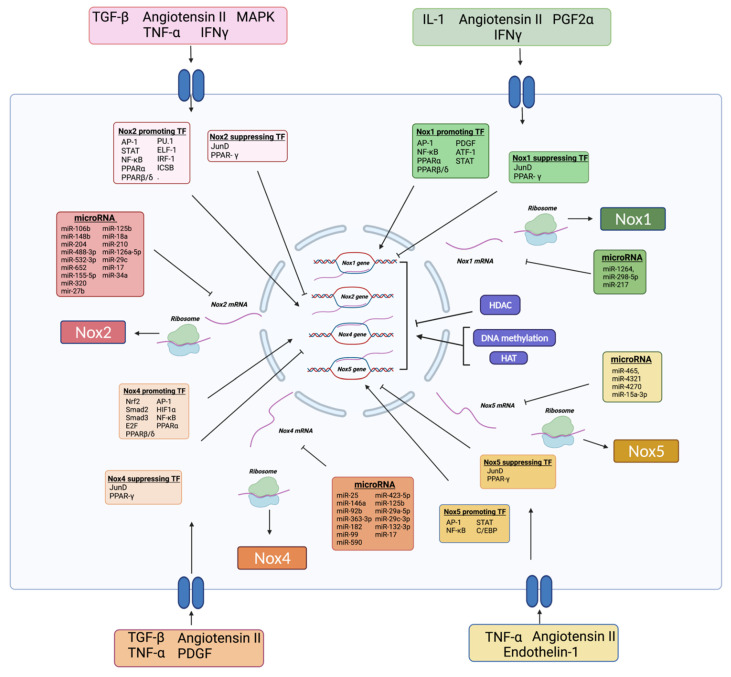
Transcriptional and epigenetic regulation of Nox. A variety of protein kinases, hormones and cytokines act to modulate transcription of Nox through Nox promoting and suppressing transcription factors. Epigenetic mechanisms play a major role regulating Nox. Within the nucleus, HDACs de-acetylate histone proteins, reducing chromatin relaxation and suppressing Nox transcription. In addition, histone acetylation (HAT) and DNA methylation increase Nox activity. Following gene transcription, Nox mRNA is degraded by a wide variety of miRs before being translated into Nox protein within ribosomes. Abbreviations: activator protein 1 (AP-1), activating transcription factor-1 (ATF-1), CCAAT/enhancer-binding proteins (C/EBP), E74-like factor 1 (ELF-1), histone deacetylation (HDAC), hypoxia-inducible factor 1 alpha (HIF1⍺), interferon consensus sequence binding protein (ICSB), interferon gamma (IFN-γ) Interferon regulatory factor-1 (IRF-1) messenger RNA (mRNA), microRNA (miR), mitogen-activated protein kinase (MAPK), Nox1 isoform of NADPH oxidase (Nox1), Nox2 isoform of NADPH oxidase (Nox2), Nox4 isoform of NADPH oxidase (Nox4), Nox5 isoform of NADPH oxidase (Nox5), nuclear factor kappa-light-chain-enhancer of activated B cells (NF-κB), peroxisome proliferator-activated receptor (PPAR), platelet derived growth factor (PDGF), prostaglandin F2α (PGF2α), signal transducer and activator of transcription protein (STAT), transforming growth factor-β (TGF-β), transcription factor (TF), tumour necrosis factor ⍺ (TNF-⍺). Created with BioRender.com (accessed on 31 August 2021).

**Figure 3 cells-10-02315-f003:**
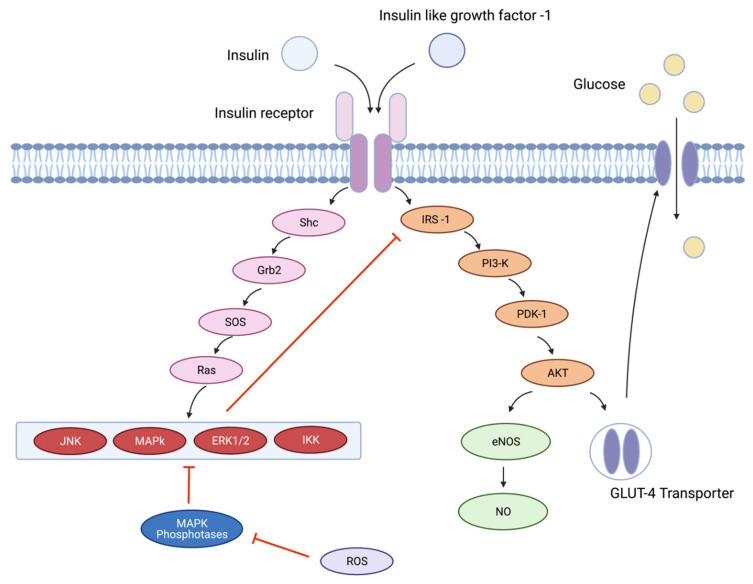
Insulin/Insulin like growth factor-1 signaling and insulin resistance. Insulin and insulin like growth factor-1 bind to the insulin receptor at the phospholipid bilayer of the cell resulting in activation of in several simultaneous protein cascades: Activation of insulin-receptor substrate-1 and downstream signaling via the interrupt the PI3K-Akt pathway, culminates in phosphorylation of Akt Phosphorylated Akt increases expression of eNOS leading to production of nitric oxide and brings about translocation of GLUT-4 transporters from the cytoplasm to the cell membrane allowing glucose to enter the cell. Activation of Shc and subsequent downstream signaling results in activation of ERK1/2, IKK, JNK and MAPK which phosphorylate IRS-1 and interrupt the PI3K-Akt pathway. Interruption of the PI3K-Akt pathway results in reduced entry of glucose into the cell and endothelial dysfunction. JNK, MAPK, ERK1/2 and IKK are inactivated by MAPK phosphates which is itself inhibited by ROS. Abbreviations: glucose transporter type 4 (GLUT-4), endothelial nitric oxide synthase (eNOS), extracellular signaling related kinase 1 and 2 (ERK1/2) growth factor receptor-bound protein 2 (Grb2), IκB kinase (IKK) Insulin receptor substrate 1 (IRS-1), c-Jun N-terminal (JNK), mitogen-activated protein kinase (MAPK), nitric oxide (NO), phosphoinositide 3-kinase (PI3K), phosphoinositide-dependent kinase-1 (PDK-1), reactive oxygen species (ROS), Son of Sevenless (SOS). Created with BioRender.com (accessed on 31 August 2021).

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
