# Peer review of "Nicotinamide Adenine Dinucleotide Phosphate Oxidases in Glucose Homeostasis and Diabetes-Related Endothelial Cell Dysfunction"

_cells, 2021, doi:10.3390/cells10092315_

Round 1
Reviewer 1 Report
The authors conducted the review article to describe the role of NOX2 and NOX4 in glucose homeostasis, endothelial function, and oxidative stress, focusing on how they are regulated in health and dysregulated in type 2 diabetes mellitus. They concluded that careful thought is needed when devising new treatments targeting NOXs and the downstream effects. However, this reviewer has several concerns regarding the contents.
(Major Comments)
Previous studies showed the endothelium-dependent relaxation’s impairment and increased release of vasoconstrictor prostanoids in arteries from animals and humans (Free Radic Biol Med 1994; 16: 383-; J Pharmacol Exp Ther 2005; 314: 1300-). Prostaglandin (PG) H2/thromboxane A2 receptor or cyclooxygenase blockade, or superoxide dismutase restores the impairment, indicating that the PGH2, cyclooxygenase, and superoxide generated contributes to the abnormality (Free Radic Biol Med 1994; 16: 383-; J Pharmacol Exp Ther 2005; 314: 1300-). Therefore, the authors must add the description regarding the role of prostanoids in the endothelial oxidative stress in the manuscript (lines 61-94).
Human aortic endothelial cells exposed to hyperglycemia showed increased expression of NOX1, oxidative stress, and proinflammatory markers in a NOX1-siRNA reversible manner (Circulation 2013; 127: 1888-). Thus, the study suggests that NOX1-dependent oxidative stress is a promising target for diabetic vasculopathy, including atherosclerosis. Also, in bovine retinal endothelial cells, which express NOX1, NOX4, and NOX5, NOX1/4 inhibition, and NOX5 silencing RNA reduced the high glucose-induced upregulation of oxidative stress (Hypertension 2020; 75: 1091-). These data indicate the potential of NOX1, NOX4, and NOX5 inhibition to reduce vision-threatening damage to the retinal endothelial cells. More importantly, NOX5 is present in endothelial cells of higher mammals, including humans, indicating that the NOX5 may be a target for pharmacological interventions in humans (J. Am. Heart Assoc. 2018, 7, e009388). Therefore, the authors have to explain the role of NOX1 and NOX5 in the manuscript additionally. Accordingly, the “NOX2 and NOX4” should be deleted in the title.
This reviewer would like to have several NOX inhibitors available at this point in the manuscript.
(Minor Comments)
“l-arginine” at line 69 must be “L-arginine.”
Reviewer 2 Report
1. The role of MGO-AGE precursor on Nox2 and 4 can be reviewed and discussed. Figure 1, a summary figure of all Nox subunits and isoforms will be helpful. In the first few paragraphs, more expanded discussion of hyperglycemia, ROS and endothelial dysfunction are needed.
2. Figure 2, any reports on lncRNA, DNA methylation and histone methylation on Nox2/4 expression and activity? Figure 3, ERK downstream of Insulin signaling was no depicted.
3. The conlusion section is too simple, please expand to give more future insights or future research direction in Nox and based therapies. "Nox4 generates H2O2 – a key signalling molecule within the endothelium"-what's next, the sentence was incomplete. Also, what's the therapeutic potential of Nox-based antioxidants compared with classical ones, such as Vitamin E, probucol, Mit-tempo, Coenzyme Q etc?
Round 2
Reviewer 1 Report
This reviewer would congratulate the excellent revision by the authors and their effort to accomplish it. However, my concern is one point as follows.
Please add the Nox5-marking on the orange symbol in Figure 1.
“The” in line 229 must be “the.”
Author Response
Thank you for your swift, thorough reviews and highlighting the above errors, they have been corrected.
Reviewer 2 Report
Most concerns have been addressed, several minor issue exist still:
- Figure 1, Nox5 not depicted.
- PGi2, page 2, is a typo.
- Previous comments for conclusion section is not revised, sentence incomplete, no future insight.
Author Response
Thank you for your swift, thorough reviews and highlighting the above errors, they have been corrected. We have also changed the sentence structure of the sentence you mention in the conclusion and added some additional points within the conclusion to address previous comments.